# Frailty trajectory and its associated factors in older patients undergoing abdominal surgery involving the digestive system: A longitudinal study

Jing Guo[1☯‡], Wenshuang Wang[2☯‡], Xiaoxue Zhang[1], Yulin Zheng[1], Xinran Wang [1]*

**1** Department of General Surgery, Xuanwu Hospital, Capital Medical University, Beijing, China, **2** Nursing Department, Beijing Tiantan Hospital, Capital Medical University, Beijing, China

☯ These authors contributed equally to this work.
‡ GJ and WWS contributed equally to this work and share first authorship.
* xwgsd2017@hotmail.com

## Abstract

Frailty is a common multifactorial clinical syndrome in older patients that seriously affects their prognosis. However, most studies to date have ignored the dynamics of frailty. Therefore, we employed a one-month observational longitudinal study to explore frailty trajectories using a latent class growth model. In total, 155 older patients who underwent abdominal surgery involving the digestive system were assessed preoperatively, at discharge, and at the one-month follow-up, and multiple logistic regression analysis was conducted to identify factors influencing frailty trajectories. Four frailty trajectory patterns were identified: no frailty (13.5%), frailty exacerbation (40.0%), frailty improvement (20.0%), and persistent frailty (26.5%). Logistic regression analysis revealed that body mass index, the Charlson comorbidity index score, the type of surgery, the intraoperative drainage tube retention time (drainage time), the first time the patient got out of bed after surgery, the time of the first oral feed after surgery, postoperative complications, mobility, nutritional risk, and anxiety were associated with frailty trajectories. We identified four frailty trajectories in older patients undergoing abdominal surgery involving the digestive system and found that these trajectories were influenced by multiple factors. Focusing on individual specificity is conducive to accurately addressing frailty-associated clinical problems and guiding relevant nursing decisions.

## 1. Introduction

With the continued growth of the aging population, the issue of health care for older adults has become a major concern [1]. Older age is no longer a limitation for surgery owing to advances in surgical technology and an increase in patient demand. Approximately 313 million operations are performed every year worldwide, 45.6% to 52.0%

**Data availability statement:** All relevant data are within the manuscript and its Supporting Information files.

**Funding:** The author(s) received no specific funding for this work.

**Competing interests:** The authors have declared that no competing interests exist.

of which are performed on older patients, with approximately one-third of these involving abdominal surgery of the digestive system [2]. Abdominal surgery of the digestive system involves the gastrointestinal tract, liver, gallbladder, and pancreas, for which substantial medical resources are needed to support the associated trauma and physiological insult. Additionally, abdominal surgery of the digestive system is often accompanied by serious complications and even death [3]. As people age, their ability to self-recover after physical damage and their capacity to respond and adapt to various external stimuli decrease, making them more prone to organic disorders of neoplastic or degenerative origin in the digestive and immune systems; this increases the risk of surgery and renders older adults more prone to perioperative adverse events, resulting in frailty-related symptoms such as activity limitations, nutritional impairment, delirium and anxiety, ultimately affecting disease prognosis [4].

Frailty refers to a decrease in individual strength, endurance, and physiological function owing to a reduction in the physiological reserves of multiple organs and systems and/or the accumulation of health defects. Negative health-related events are more likely to occur under stress; however, these events can be prevented or delayed with appropriate interventions [5]. Frailty is a common problem in older people, especially surgical patients. The mechanisms involved in the biology of frailty are not clear and may be influenced by a variety of factors, such as age, body mass index (BMI), education level, comorbidity, nutritional status, activity capacity, environment, and mental state [6]. Kumar et al. [7] reported that, among 10,000 older inpatients in China, the average prevalence of frailty was 37% in those over 60 years of age who underwent general surgery and as high as 65% among older patients who underwent cardiac, thoracic, and abdominal surgery. Although many studies have investigated frailty and its related factors in older patients, most have been cross-sectional [8,9]. Frailty is strongly associated with poor clinical outcomes. According to the National Surgical Quality Improvement Program (NSQIP) database, frail older patients have a higher incidence of postoperative complications during hospitalization and a twofold greater risk of unplanned readmission within 30 days after surgery than do nonfrail patients [10]. Frailty leads to serious postoperative complications, disability, unplanned readmission, increased long-term care needs, decreased quality of life, and even death, which increases medical and social burdens. It is expected to continue to be a major challenge in clinical medicine and nursing [11].

Frailty accurately reflects the health problems and medical needs of older people and is an important indicator for patient risk assessment. Moreover, given its dynamic nature, frailty is also a reversible condition. Thus, closely monitoring the trajectory of frailty in older surgical patients may improve their prognosis during the perioperative period and support clinical decision-making. Several studies have examined the trajectory of frailty in older patients. A preliminary investigation [12] revealed that the frailty rate of patients with gastric cancer was significantly greater after surgery than before surgery and began to decrease after one month. Iwakura et al. [13] investigated frailty after kidney transplantation and reported that the incidence of frailty increased in the first month after transplantation and returned to the preoperative level in the second month. Compared with that in kidney

transplant patients, the decrease in the incidence of frailty in gastric cancer patients was not apparent one month after surgery [14]. The frailty trajectories of older patients depend on various factors and are heterogeneous. Wang et al. [15] reported that frailty in older community-dwelling patients with moderate to severe hypertension exhibited four progression trajectory types—"low-stable", "low-rapid growth", "moderate-rapid growth", and "high-slow growth". In a longitudinal study, Jung et al. [16] identified three trajectories of frailty development in older inpatients, namely, "low-stable", "low-rapid growth", and "high level". However, no study to date has explored the development trajectory of frailty in older patients undergoing abdominal surgery involving the digestive system or the factors that influence the different trajectory categories.

In this study, we aimed to answer three questions: First, does frailty change in older patients undergoing abdominal surgery of the digestive system in the short term? Second, are there different trajectories of frailty development among older patients undergoing abdominal surgery involving the digestive system? Third, what factors influence the development trajectory of the different types of frailty?

## 2. Materials and methods

### 2.1. Participants

This study was conducted at a Grade-A tertiary hospital in Beijing, China. Participant recruitment commenced on June 1, 2023, and continued through July 31, 2024. The inclusion criteria for this study were as follows: (1) aged ≥60 years; (2) underwent abdominal surgery of the digestive system, such as cholecystectomy bile duct/channel stone removal, hepatectomy, radical colorectal surgery, pancreatic necrosectomy, partial gastrectomy, or any surgical approach; (3) had sufficient cognition and hearing to answer questions correctly; and (4) were willing to participate in the study. The exclusion criteria were: (1) receiving emergency surgery; (2) taking antidepressants such as donepezil or levodopa (which have been shown to cause symptoms similar to fatigue) [17]; and (3) being unable to participate in postdischarge follow-up. Data for patients who voluntarily quit, did not cooperate during follow-up, were lost to follow-up, whose disease condition changed, or died were not included in the study.

Once the patients' operation times and methods had been determined (preoperation, T1), the patients were recruited via convenience sampling. Patients were assessed at discharge (7–9 days after surgery, T2) and at the one-month follow-up (T3). The purpose, procedure, and necessary precautions of the study were explained to the patients before the survey, and the data were used only for the study report. The patients were free to withdraw at any time without affecting subsequent treatment, and informed consent was obtained from the patients. At baseline (T1), the general information (e.g., sex, age, BMI, education level, marital status, monthly income, pain level, and comorbidities) of the patients was collected, and the timing of the operation, surgery-related characteristics (e.g., operation name, operation grade, operation duration, intraoperative blood loss, and whether to leave the abdominal drainage tube), postoperative follow-up date, and postoperative information (e.g., whether the patient was admitted to the intensive care unit [ICU] for treatment after surgery, the time of the first oral feed and first anal exhaust after surgery, abdominal drainage tube indwelling time, and postoperative complications) were recorded.

A random effects model in repeated measurements analysis of variance was employed to calculate the sample size. Assuming a 20% loss to follow-up, 150 subjects needed to be included in the study.

### 2.2. Ethical considerations

The research was conducted according to the principles of the World Medical Association Declaration of Helsinki. All participants provided informed consent (researchers explained the purpose and methods of the study to the participants, obtained the informed consent forms, and distributed questionnaires). And the study was approved by the Medical Ethics Committee of Xuanwu Hospital, Capital Medical University, China. (Clinical Research Ethics Review [2020] No. 064).

### 2.3. Variables

**2.3.1. Frailty.** The Fried frailty scale includes 5 components—unexplained weight loss, slow walking speed, poor grip strength [18], significantly reduced activity, and self-perceived fatigue [19]. Xi was the first to introduce this scale into frailty research in older Chinese adults [20]. If three or more of these criteria are met, a patient is considered frail, and if one or two criteria are met, the patient is in a prefrail state; a patient not meeting any of these criteria is considered not frail/healthy. This scale has a pathophysiological basis, including subjective and objective indicators. The American guidelines for the preoperative evaluation of older surgical patients recommend the Fried scale as the preferred tool for frailty screening and evaluation. It is the most widely used index for the assessment of frailty.

**2.3.2. Activities of daily living (ADL).** The ADL index includes the following 10 items: eating, grooming, bathing, dressing, bowel control, toileting, bladder control, bed and chair transfer, walking, and walking up and down stairs [21]. According to whether older adults need help or the degree of help needed, they are scored via a 5-point Likert scale, with higher scores indicating stronger independence (≤ 40: severely dependent, completely dependent on others; 41–60: moderately dependent, mostly dependent on others; 61–99: slightly dependent, partially dependent on others; 100: completely independent, without the need for care).

**2.3.3. Nutrition.** The NRS 2002 includes preliminary and final screening of nutritional risk [22]. The preliminary screening included four items, which were all negative and were resurveyed after one week. The final screening consisted of one item, which was positive, and comprised three aspects. The total score of the scale was calculated by adding all the scores. The highest score was 7, and a total score ≥3 was considered a nutritional risk.

**2.3.4. Anxiety.** This scale is used to assess temporary anxiety states (state anxiety, S-AI) and trait anxiety tendencies (trait anxiety, T-AI) [23]. The first 20 questions measure state anxiety, which refers to a recent or immediate feeling, and assess anxiety in stressful situations; the latter 20 questions measure trait anxiety, which is a regular emotional experience. The STAI was translated into Chinese by Li [24]. The scale is scored from 1 to 4, with a total score of 20–80. The higher the score is, the more serious the anxiety. The test showed good reliability and validity in the Chinese sample.

### 2.4. Statistical analysis

SPSS Statistics version 26 (IBM Corp., Chicago, IL, USA) was used to calculate the scores of frailty and the measured variables. Mplus 8.3 (Muthén & Muthén, Los Angeles, CA, USA) was used for latent class growth analysis (LCGA) to explore the different frailty trajectories in older patients after major abdominal surgery [25]. When a model is built, the number of potential classes gradually increases, beginning with a one-class model. The best model was determined by combining the results of the model fitting evaluation index and the interpretability of the results, including the Akaike information criterion (AIC), Bayesian information criterion (BIC), and sample size-adjusted BIC (aBIC). The fitting performance was judged by comparing the difference between the expected and actual values. The smaller the value is, the better the fitting effect. The classification accuracy of the model was evaluated via entropy. The classification effect of a model is considered better if the value is > 0.80. The Lo–Mendell–Rubin (LMR) test and the bootstrapped likelihood ratio test (BLRT) are commonly used and sensitive test methods for model fitting. When the $P$ value of the LMR test or the BLRT is < 0.05 in $k$ categories, $k$-category models are selected in preference to $k-1$ models, and *vice versa*.

Finally, SPSS 26.0 was also used to analyze the characteristics of the different trajectories of frailty. For univariate analysis, the chi-square test was used to determine the associations between categorical variables (such as sex, education, and marital status), whereas one-way ANOVA was used to determine associations for continuous variables (such as age, self-efficacy, social support, and other factors of self-care behavior). After a test for homogeneity of variance, if the variance was uneven, a nonparametric test was used. Factors with a $P$ value <0.05 in the univariate analysis were included in the multiple logistic regression and are shown in the forest plot. A P value <0.05 was considered significant.

## 3. Results

### 3.1. Participant characteristics

Overall, 164 patients completed the baseline survey, and 478 data points were collected. Eleven patients were excluded. Finally, 155 patients completed the survey at the three time points, and 155 complete questionnaires were collected, representing a recovery rate of 93.4% (Fig 1).

Among the 155 participants surveyed at baseline, the mean age was 70.38±0.55 years, and 76 (49%) were male. The most common surgical procedure was cholecystectomy (23.2%), and the median operative time was 5.6 hours. All the operations were laparoscopic surgeries performed under general anesthesia (Table 1).

### 3.2. Frailty status

The results revealed that, before surgery (T1), 43 patients (27.7%) presented with frailty, 62 (40.0%) presented with prefrailty, and 50 (32.3%) presented with no frailty/were healthy patients. At discharge (T2), the incidence of frailty increased to 43.9%, that of prefrailty decreased to 32.2%, and 23.9% of the patients were not frail and were healthy. The frailty rate decreased to 29.0% one month after surgery (T3) but was still higher than that at T1. Patients with prefrailty accounted for 27.7% of the total, and those without frailty/who were healthy accounted for 43.2% (Table 2).

### 3.3. Frailty trajectories

In this study, latent class growth models (LCGMs) of 1–5 categories were extracted to fit the repeated measurement data. Although the 5-class model had the lowest AIC and highest entropy, the associated BLRT *P* value was 0.030. Therefore, in this study, we compared only the 4-class and 3-class models. We found that the AIC, BIC, and aBIC of the

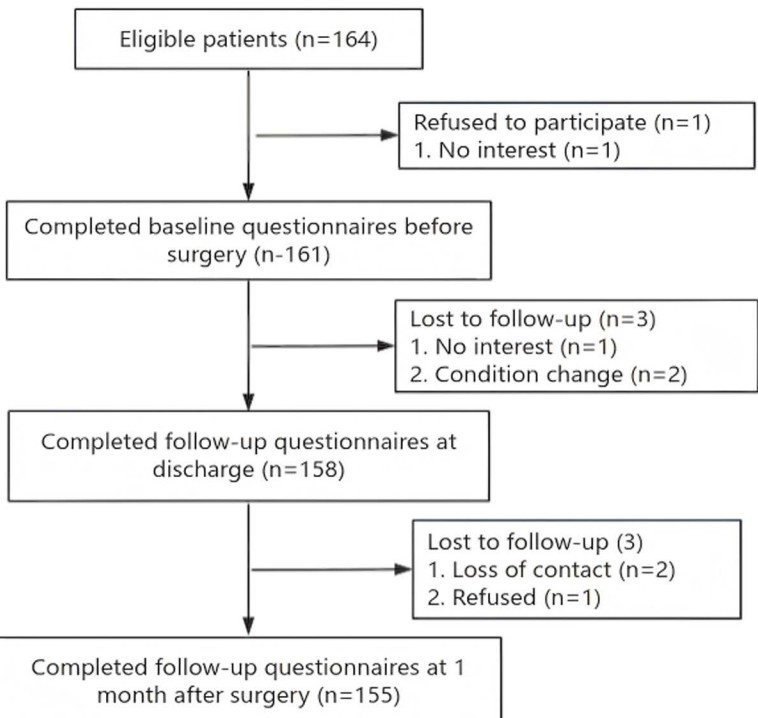

**Fig 1. Research Subject Inclusion Flowchart.**

four-category model were lower than those of the three-category model, and the *P* value of the BLRT for the four categories was significant (< 0.001), thus showing the best-fitting effect. Accordingly, the four-category model was chosen as the final model of the development of and change in frailty.

Four types of frailty trajectories were identified in older patients after major abdominal surgery. Class 1 comprised 31 patients, accounting for 20.1% of the total. These patients were not frail or prefrail before or after surgery, and their state was stable within one month postsurgery; this group of patients was named the "nonfrail group". Class 2 consisted of 62

**Table 1. Baseline Demographic and Clinical Characteristics of Study Participants.**

| Variables | | n(%)or Mean±SD | Variable | | n(%)or Mean±SD |
|---|---|---|---|---|---|
| Gender | Male | 76(49.0) | CCI | Severe (≥ 6 points) | 126(81.3) |
| | Female | 79(51.0) | Surgical Approach | Laparotomy | 52(33.5) |
| Age(years) | 60~74 | 113(73.0) | | Laparoscopy | 103(66.5) |
| | 75~89 | 42(27.0) | Anesthesia Type | General anesthesia | 155(100.0) |
| BMI (kg/m²) | Underweight (<18.5) | 5(3.2) | Surgical Site | Liver/ Pancreas | 44(28.4) |
| | Normal (18.5–23.9) | 67(43.2) | | Colorectal | 40(25.8) |
| | Overweight (24–27.9) | 44(28.4) | | Upper gastrointesinal | 71(45.8) |
| | Obese (≥28) | 39(25.2) | Surgery Duration (hours) | | 3.17±0.19* |
| Smoking Status | Never smoked | 117(75.5) | Preoperative nasogastric tube | NO | 108(69.7) |
| | Current smoker | 19(12.3) | | Yes | 47(30.3) |
| | Former smoker | 19(12.3) | Preoperative urinary catheter | NO | 136(87.7) |
| Alcohol Use | Never used | 113(72.9) | | Yes | 19(12.3) |
| | Current user | 31(20) | Abdominal Drainage Tube | NO | 65(41.9) |
| | Former user | 11(7.1) | | Yes | 90(58.1) |
| Marital Status | Married | 130(83.9) | Indwelling Time of Abdominal Drainage Tube(days) | | 3.61±0.50* |
| | Divorced or widowed | 25(16.1) | ICU Admission | No | 107(69) |
| Education Level | Elementary school or below | 34(21.9) | | Yes | 48(31) |
| | Junior high school | 96(61.9) | ICU Stay(days) | | 1.61±0.31* |
| | High school/college | 7(4.5) | Postoperative Pain (24h) | No (0) | 0 |
| | University or above | 18(11.6) | | Mild (1~3 points) | 4(2.6) |
| Monthly Income (CNY) | <1,000 | 0 | | Moderate (4~6 points) | 104(67.1) |
| | 1,000~3,000 | 2(1.3) | | Severe (7~10 points) | 47(30.3) |
| | 3,000~5,000 | 9(5.8) | Postoperative Analgesia | No | 67(43.2) |
| | >5,000 | 144(92.9) | | Yes | 88(56.8) |
| Medical Insurance | Rural resident insurance | 11(7.1) | First Ambulation (days) | | 2.75±0.26* |
| | Employee health insurance | 6(3.9) | First Anal Exhaust (days) | | 2.02±0.12* |
| | Public expense insurance | 80(51.6) | First Oral Feeding (days) | | 2.83±0.94* |
| | Self-paid | 58(37.4) | Postoperative Complications | No | 57(36.8) |
| CCI | Low (2~3 points) | 8(5.2) | | YES | 98(63.2) |
| | Moderate (4~5 points) | 21(13.5) | Length of Stay (days) | | 8.36±2.177* |

Abbreviations: *:mean±SD; BMI: Body Mass Index; CCI: Charlson Comorbidity Index; ICU: Intensive Care Unit.

Smoking Status:Self-reported alcohol consumption history;Monthly Income (CNY):Self-reported monthly income before surgery; Preoperative nasogastric tube:Has a nasogastric tube been placed before surgery?Preoperative urinary catheter: Has a urinary catheter been placed before surgery? Postoperative Pain (24h) the patient's pain level within 24 hours after surgeryNumeric Rating Scale (0–10): 0 = no pain; 1–3 = mild; 4–6 = moderate; 7–10 = severe;First Ambulation/ Anal Exhaust/Oral Feeding(days):Time to first out-of-bed activity/anal exhaus/oral intake postoperatively.

patients (39.6%) who were not in a state of frailty or prefrailty before the operation (T1) but whose state worsened after surgery, newly manifesting as frailty; this state continued for one month after the operation, without effective relief. Compared with the preoperative state (T1), the frailty state of patients in this category was significantly aggravated when they were discharged from the hospital (T2); these patients were categorized into the "frailty exacerbation group". There were 41 patients in class 3, accounting for 26.7% of the total. These patients were frail before the operation (T1), and the frailty state lasted until one month after the operation (T3); these patients were placed in the "persistent frailty group". Class 4 comprised 21 patients and accounted for 13.6% of the total patients. Patients in this category were in a state of frailty or prefrailty before the operation (T1), but their status continuously improved postoperatively and could revert to a nonfrail/ healthy state one month after surgery; patients in this group were classified into the "frailty improvement group" (Fig 2).

### 3.4. The effect of longitudinal changes in variables on frailty trajectories

In the univariate analysis, age, BMI, education level, the Charlson Comorbidity Index (CCI) score, surgery type, the presence of a gastric tube preoperatively, the presence of a urinary catheter, intraoperative abdominal drainage tube indwelling time, time spent in the ICU, receiving postoperative analgesia/sedation drugs, first postoperative ambulation time, time of the first postoperative oral feed, postoperative complications, daily activity ability, nutritional risk, and anxiety were significantly different among the four trajectories ($P<0.05$) (Tables 3–5). Factors with significant $P$ values in the univariate analysis were included in the multiple logistic regression analysis.

**Table 2. Frailty Status Trajectory (T1–T3).**

| Frailty Status | T1 (Preoperative) | T2 (7–9 Days Postoperative) | T3 (1 Month Postoperative) | $x^2$ | *P*-value |
|---|---|---|---|---|---|
| Nonfrail | 50(32.3) | 37(23.9) | 67(43.2 | 16.175 | 0.001 |
| Prefrailty | 62 (40.0) | 50 (32.2) | 43 (27.7) | | |
| Frailty | 43(27.7) | 68 (43.9) | 45 (29.0) | | |

Abbreviations: T1: preoperation; T2: 7–9 days after surgery; T3: one month after surgery.

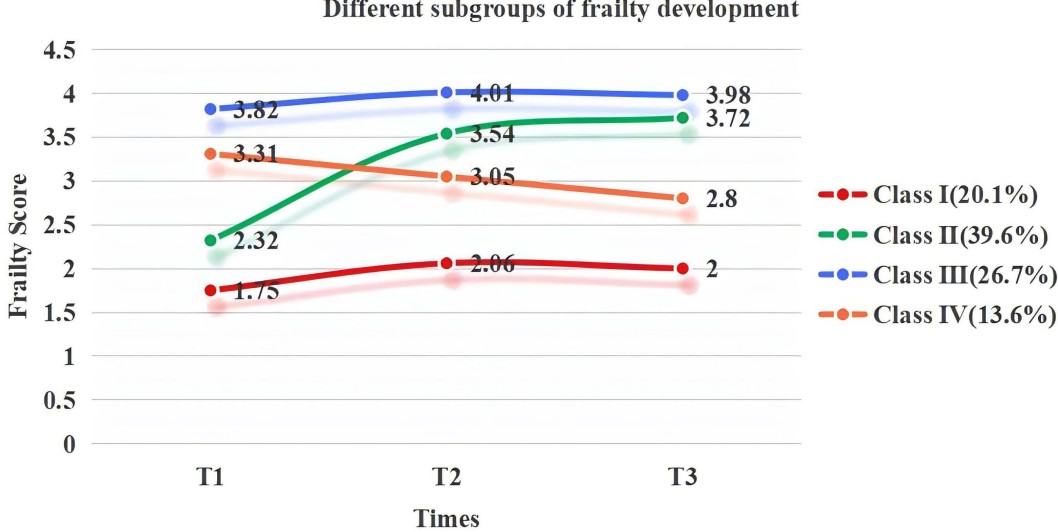

**Fig 2. Different subgroups of frailty development.**

**Table 3. Univariate Analysis of Participant Characteristics by Frailty Trajectory Group.**

| Variable | | No nfrail group (n=21) | Frailty exacerbation group (n=62) | Persistent frailty group (n=41) | Frailty improvement group (n=31) | χ² | P-value |
|---|---|---|---|---|---|---|---|
| Gender | Male | 9 (47.6) | 32 (51) | 19 (46.7) | 16 (51.6) | 5.905 | 0.052 |
| | Female | 11 (52.4) | 30 (49) | 22 (53.3) | 15 (48.4) | 20.587 | **0.001** |
| Age (years) | 60~74 years | 17 (78.6) | 40 (65.3) | 29 (70.7) | 27 (91.2) | | |
| | 75~89 years | 4 (21.4) | 22 (34.7) | 12 (29.3) | 4 (8.8) | | |
| BMI (kg/m²) | Underweight (<18.5) | 1 (4.8) | 2 (3.2) | 1 (3.3) | 1 (2.9) | 23.965* | **<0.001** |
| | Normal (18.5–23.9) | 17 (81) | 19 (30.6) | 15 (36.7) | 16 (55.9) | | |
| | Overweight (24–27.9) | 2 (11.9) | 16 (25.8) | 14 (33.3) | 12 (32.4) | | |
| | Obese (≥28) | 1 (2.4) | 25 (40.3) | 11 (26.7) | 2 (8.8) | | |
| Smoking Status | Never smoked | 13 (54.8) | 48 (77.4) | 30 (73.3) | 23 (76.5) | 4.683* | 0.321 |
| | Current smoker | 4 (11.9) | 7 (12.2) | 5 (13.3) | 4 (11.8) | | |
| | Former smoker | 4 (11.9) | 7 (12.2) | 6 (14.6) | 4 (11.8) | | |
| Alcohol Use | Never used | 15 (73.8) | 46 (74.2) | 30 (73.3) | 22 (73.5) | 4.581* | 0.333 |
| | Current user | 4 (19) | 13 (21.0) | 8 (20) | 6 (20.6) | | |
| | Former user | 2 (7.1) | 3 (4.8) | 3 (6.7) | 3 (5.9) | | |
| Marital Status | Married | 16 (76.2) | 53 (85.5) | 35 (85.4) | 26 (83.9) | 8.979* | 0.062 |
| | Not married (divorced or widowed) | 5 (23.8) | 9 (14.5) | 6 (14.6) | 5 (16.1) | | |
| Education Level | Elementary school and below | 0 | 16 (25.8) | 17 (41.5) | 1 (3.2) | 13.237* | 0.039 |
| | Junior high school | 15 (71.4) | 41 (66.1) | 21 (51.2) | 19 (61.3) | | |
| | High school or college | 1 (4.8) | 5 (8.1) | 1 (2.4) | 0 | | |
| | University and above | 5 (2.4) | 0 | 2 (4.9) | 11 (35.5) | | |
| Monthly Income(CNY) | <1,000 | 0 | 0 | 0 | 0 | 4.324* | 0.633 |
| | 1,000~3,000 | 0 | 2 (3.2) | 0 | 0 | | |
| | 3,000~5,000 | 0 | 4 (6.5) | 4 (9.8) | 1 (3.2) | | |
| | >5,000 | 21 (100.0) | 56 (90.3) | 37 (90.2) | 30 (96.8) | | |
| Medical Insurance | Rural resident insurance | 3 (14.3) | 1 (1.6) | 5 (12.2) | 2 (15.4) | 6.199* | 0.401 |
| | Employee health insurance | 1 (4.8) | 2 (3.2) | 3 (7.3) | 0 | | |
| | Public expense insurance | 13 (61.9) | 41 (66.1) | 20 (48.8) | 6 (119.4) | | |
| | Self-paid | 4 (2.4) | 18 (29.0) | 13 (31.7) | 23 (74.2) | | |
| CCI | Low (2~3 points) | 5 (23.8) | 0 | 0 | 3 (9.7) | 55.629* | **0.01** |
| | Moderate (4~5 points) | 10 (47.6) | 5 (8.1) | 3 (7.3) | 3 (9.7) | | |
| | Severe (≥ 6 points) | 6 (28.6) | 57 (91.9) | 38 (92.7) | 25 (80.6) | | |

*, Fisher's exact test; BMI: Body Mass Index; CCI: Charlson Comorbidity Index.

**Table 4. Univariate Analysis of Clinical Characteristics by Frailty Trajectory Group.**

| Variable | | No frailty group (*n*=21) | Frailty exacerbation group (*n*=62) | Persistent frailty group (*n*=41) | Frailty improvement group (*n*=31) | $F/\chi^2$ | *P*-value |
|---|---|---|---|---|---|---|---|
| Surgical Approach | Laparotomy | 1(4.8) | 39(62.9) | 12(29.3) | 0 | | |
| | Laparoscopy | 20(95.2) | 23(37.1) | 29(70.7) | 31(100.0) | | |
| Anesthesia Type | General anesthesia | 21(100.0) | 62(100.0) | 41(100.0) | 31(100.0) | – | – |
| Surgical Site | Liver/Pancreas | 2(9.5) | 23 (37.1) | 18(43.9) | 1(3.2) | 21.204* | **0.002** |
| | Colorectal | 4(19.0) | 18 (29.0) | 15(36.6) | 3(9.7) | | |
| | Upper gastrointesinal | 15(71.4) | 21 (33.9) | 8(19.5) | 27(87.1) | | |
| Surgery Duration (hours) | | 1.02±0.02 | 3.25±0.21 | 3.55±0.42 | 1.31±0.06 | 1.524* | 0.158 |
| Preoperative naso-gastric tube | NO | 20 (95.2) | 37 (59.2) | 21(51.2) | 30 (97.1) | 10.309 | **0.006** |
| | Yes | 1 (4.8) | 25 (40.8) | 20(48.8) | 1 (2.9) | | |
| Preoperative uri-nary catheter | NO | 21 (100) | 48 (77.6) | 36(87.8) | 31 (100) | 8.461 | **0.015** |
| | Yes | 0 | 14 (22.4) | 5 (12.2) | 0 | | |
| Abdominal drainage tube | NO | 19 (88.1) | 19 (30.6) | 1 (3.3) | 26 (82.3) | 38.351 | **<0.001** |
| | Yes | 2 (11.9) | 43 (69.4) | 40 (96.7) | 5 (17.6) | | |
| Indwelling Time of Abdominal Drainage Tube(days) | | 1.08±0.02 | 3.25±0.65 | 3.57±0.59 | 1.59±0.08 | 41.423* | 0.057 |
| ICU Admission | No | 20 (97.6) | 30 (49) | 27(65.9) | 30 (97.1) | 53.516 | **<0.001** |
| | Yes | 1 (2.4) | 32 (51) | 14(34.1) | 1 (2.9) | | |
| ICU Stay(days) | | 0.95±0.02 | 2.85±0.12 | 3.11±0.64 | 1.07±0.05 | 54.084* | **<0.001** |
| Postoperative Pain (24h) | No (0) | 0 | 0 | 0 | 0 | 4.462 | 0.064 |
| | Mild (1~3 points) | 3(14.3) | 0 | 0 | 1(3.2) | | |
| | Moderate (4~6 points) | 18(85.7) | 34(54.8) | 22(53.7) | 30(96.8) | | |
| | Severe (7~10 points) | 0 | 28(45.2) | 19(46.3) | 0 | | |
| Postoperative Analgesia | No | 19 (90.5) | 13 (20.4) | 7 (16.7) | 28(90.3) | 36.616 | **<0.001** |
| | Yes | 2 (9.5) | 49 (79.6) | 34 (83.3) | 3 (9.7) | | |
| First Ambulation (days) | | 1.85±0.02 | 3.76±0.58 | 3.96±0.53 | 2.05±0.47 | 5.723* | **<0.001** |
| First Anal Exhaust (days) | | 1.52±0.05 | 2.38±0.67 | 2.59±0.83 | 1.87±0.42 | 2.694* | 0.408 |
| First Oral Feeding (days) | | 1.94±0.13 | 3.85±0.68 | 3.80±0.95 | 2.35±0.67 | 55.245* | **<0.001** |
| Length of Stay (days) | | 7.10±1.05 | 8.37±2.15 | 9.03±3.09 | 7.26±1.48 | 7.238* | 0.057 |
| Postoperative complications | No | 20(95.2) | 5(8.1) | 4(9.8) | 28(90.3) | 67.638 | **<0.001** |
| | YES | 1(4.8) | 57(91.2) | 37(89.2) | 3(8.7) | | |

Abbreviations: *: repeated measures ANOVA; ICU: Intensive Care Unit.

Compared with those in the nonfrail group, BMI, postoperative complications, activity capacity, and nutritional risk were greater in the frailty exacerbation group; thus, these factors had a significant effect on this group (Fig 3). The factors influencing the persistent frailty group were BMI, drainage time, first postoperative ambulation time, nutritional risk, and anxiety (Fig 4). Compared with the persistent frailty group, BMI, surgery type, first postoperative ambulation time, time to first postoperative oral feed, and activity ability had greater impacts on patients in the frailty improvement group (Fig 5).

**Table 5. Univariate Analysis of Assessment-related Characteristics by Frailty Trajectory Group.**

| Variable | | No frailty group (*n*=21) | Frailty exacerbation group (*n*=62) | Frailty improvement group (*n*=31) | persistent frailty group (*n*=41) | *F* | *P*-value |
|---|---|---|---|---|---|---|---|
| ADL | | 96.72±0.530 | 93.56±0.648 | 94.53±0.819 | 91.03±1.702 | 32.019 | **<0.001** |
| NRS 2002 | | 2.08±0.039 | 2.18±0.073 | 3.04±0.983 | 3.34±1.609 | 11.626 | **<0.001** |
| Anxiety | S-AI | 29.38±0.753 | 31.48±1.024 | 28.76±0.507 | 32.24±0.427 | 8.442 | **0.047** |
| | T-AI | 34.47±0.938 | 42.55±0.798 | 30.65±1.278 | 43.16±0.498 | 7.298 | **0.001** |

Abbreviations: ADL, activities of daily living; NRS 2002, nutritional risk screening 2002; STAI, state-trait anxiety inventory; S-AI, state anxiety; T-AI, trait anxiety.

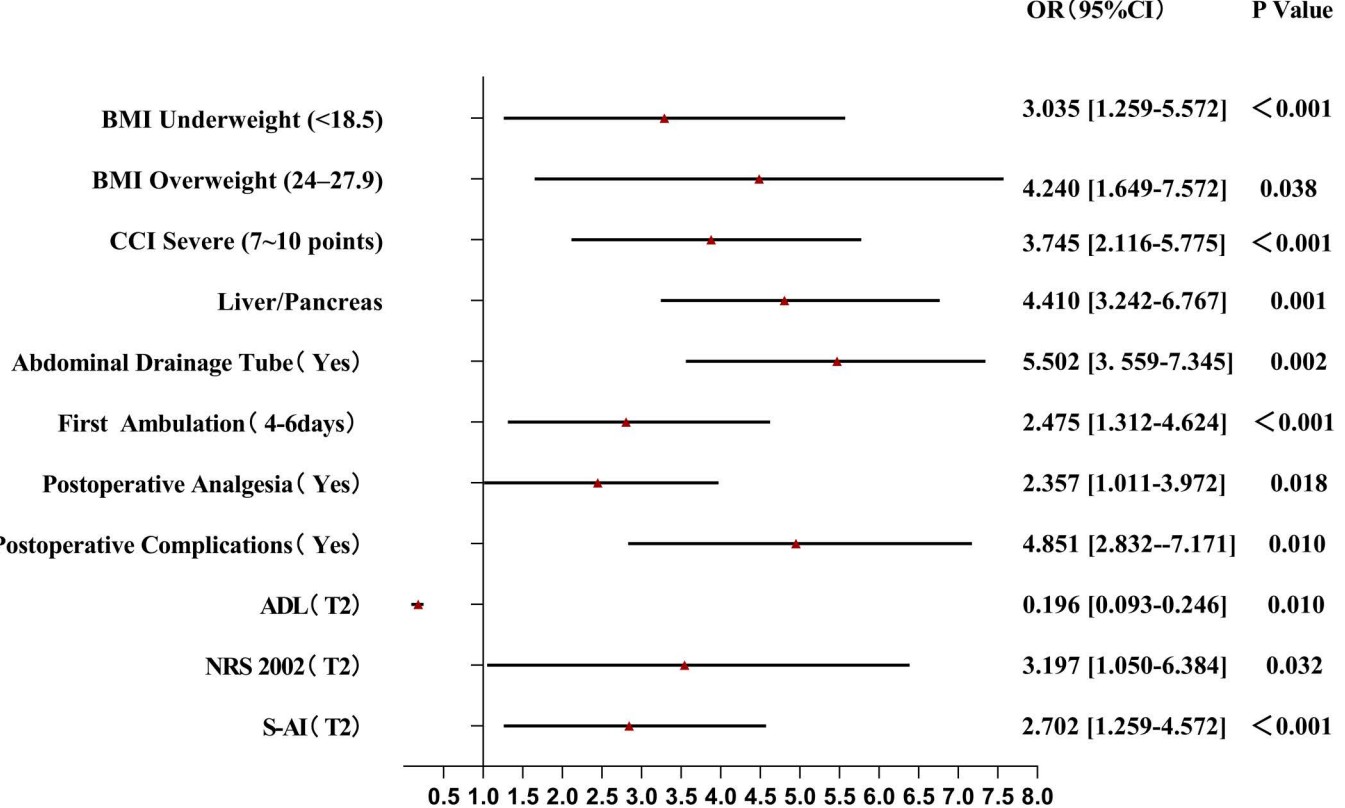

Fig 3. Forest Plot of Multinomial Logistic Regression Analysis for Frailty Exacerbation group.

## 4. Discussion

### 4.1. Frailty status in older patients undergoing abdominal surgery involving the digestive system

In total, 21.3% of the older patients who underwent abdominal surgery of the digestive system and who were included in this study were frail before surgery, which was similar to the findings of Yang et al. [26]. Patients who undergo abdominal surgery of the digestive system are in a severe condition and present with a persistent decline in multiple organ system functions. We found that, compared with preoperative patients, patients were significantly frailer at discharge, and the overall incidence of frailty remained higher one month after surgery than before the operation. Patients are in a state of

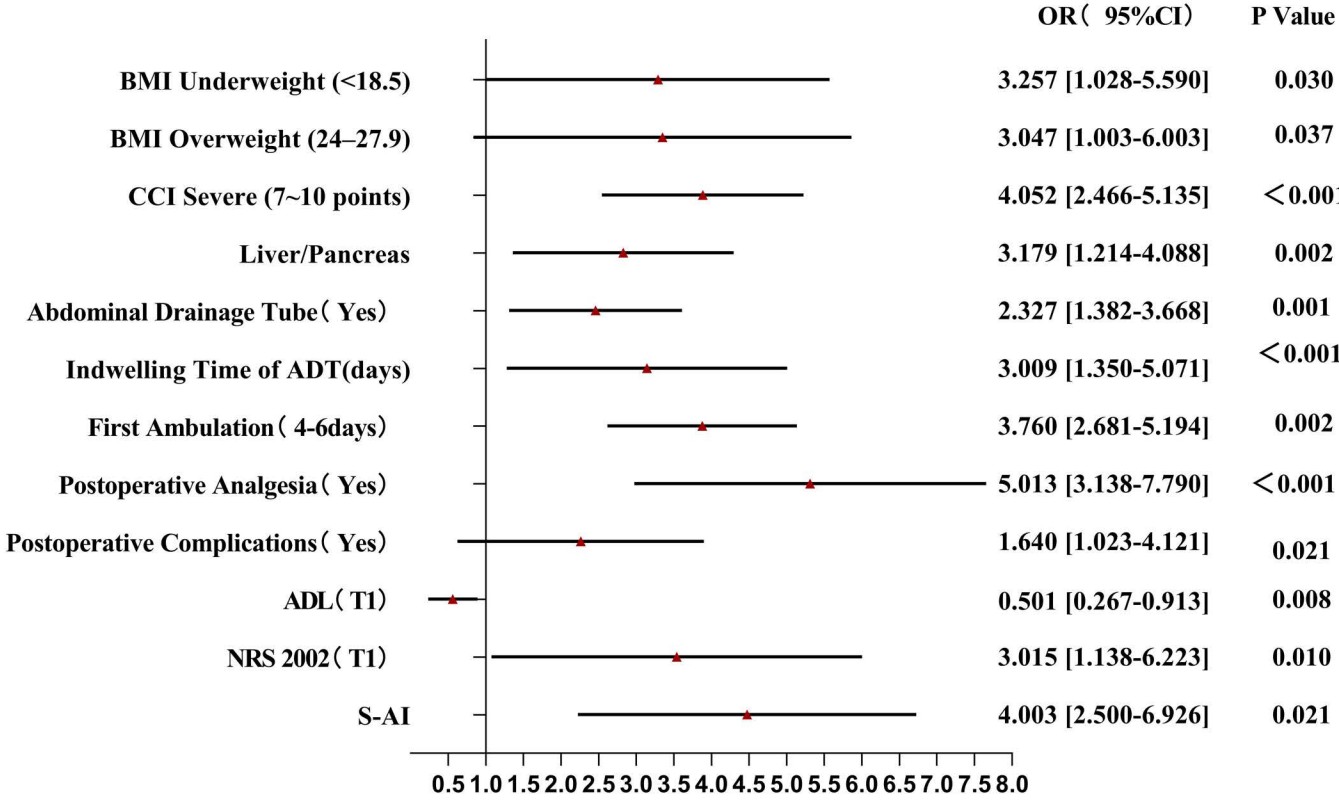

| | OR（95%CI） | P Value |
|---|---|---|
| BMI Underweight (<18.5) | 3.257 [1.028-5.590] | 0.030 |
| BMI Overweight (24–27.9) | 3.047 [1.003-6.003] | 0.037 |
| CCI Severe (7~10 points) | 4.052 [2.466-5.135] | ＜0.001 |
| Liver/Pancreas | 3.179 [1.214-4.088] | 0.002 |
| Abdominal Drainage Tube（Yes） | 2.327 [1.382-3.668] | 0.001 |
| Indwelling Time of ADT(days) | 3.009 [1.350-5.071] | ＜0.001 |
| First Ambulation（4-6days） | 3.760 [2.681-5.194] | 0.002 |
| Postoperative Analgesia（Yes） | 5.013 [3.138-7.790] | ＜0.001 |
| Postoperative Complications（Yes） | 1.640 [1.023-4.121] | 0.021 |
| ADL（T1） | 0.501 [0.267-0.913] | 0.008 |
| NRS 2002（T1） | 3.015 [1.138-6.223] | 0.010 |
| S-AI | 4.003 [2.500-6.926] | 0.021 |

ADT：Abdominal Drainage Tube

**Fig 4. Forest Plot of Multinomial Logistic Regression Analysis for Persistent Frailty group.**

high consumption after surgery, with insufficient compensatory capacity and weakened immunity. Frailty is more likely to occur under the stimulation of therapeutic procedures and in a hospital environment. However, with the clinical application of "enhanced recovery after surgery" (ERAS) programs, the postoperative hospital stay is shortened, and medical care services are greatly reduced after discharge [27]. Patients lack disease-related knowledge and self-management ability, which also aggravates the development of frailty. The determination of frailty status is an important preoperative risk assessment for older patients.

## 4.2. Frailty trajectory in older patients undergoing abdominal surgery involving the digestive system

In this study, we fitted four frailty trajectories for older patients undergoing abdominal surgery of the digestive system and categorized them into a nonfrail group, a frailty exacerbation group, a frailty improvement group, and a persistent frailty group, thus providing further evidence for heterogeneity in frailty development. The proportion of patients in the frailty exacerbation and persistent frailty groups was relatively high, likely because older surgical patients are at greater risk of disease, their physiological functions deteriorate, their independence may be lost, and their negative emotions may increase. Additionally, postoperatively, the bodies of older patients are stimulated by inflammatory reactions and immune system disorders, as well as activity restrictions, the effects of drugs, the environment, and role changes, which often aggravate their frailty [28]. Patients in these groups need long-term follow-up and continuous care. In addition, we found that frailty may exhibit a trend of "improvement". These patients had a normal BMI, the surgical option was in the

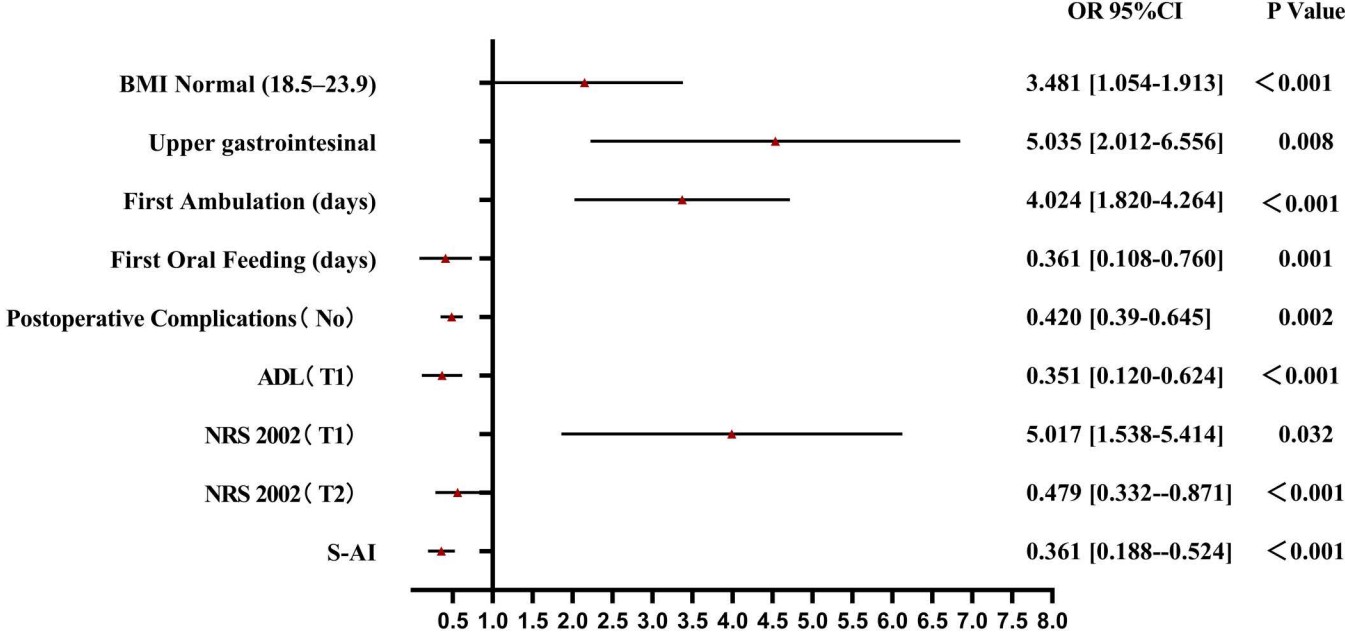

**Fig 5. Forest Plot of Multinomial Logistic Regression Analysis for Frailty Improvement group.**

gallbladder, and no postoperative complications occurred; moreover, owing to the high preoperative ADL score, low nutritional risk and S-AI risk, the patients had early oral feeding and ambulation after surgery. They had strong stress-enduring ability and good tolerance to surgical treatment and may also have had access to early postoperative rehabilitation exercise, which improved the original frail state or even reverted it to a nonfrail state. However, owing to the limitation of follow-up time in our study, the changes in frailty status were not pronounced, a shortcoming that should be addressed in future studies. The development of a frailty curve provides possibilities for the prevention, delay, and reversal of this process. Focusing on individual characteristics is conducive to attenuating frailty and provides a basis for the formulation and effective implementation of frailty intervention plans.

### 4.3. Factors influencing frailty trajectories in older patients undergoing abdominal surgery involving the digestive system

**4.3.1. Risk factors for frailty trajectory exacerbation.** The results of this study revealed that an abnormal BMI, comorbidities, upper GI surgery or pancreatic surgery, intraoperative abdominal drainage tube indwelling time, postoperative complications, and anxiety may be risk factors for frailty exacerbation in patients. An abnormal BMI, associated with malnutrition and obesity, increases the risk of frailty. On the one hand, patients who undergo abdominal surgery often have digestive problems and experience perioperative fasting, resulting in inadequate energy and protein intake, which increases the risk of frailty. On the other hand, patients with a high BMI have more adipose tissue and, consequently, secrete greater amounts of adiponectin, IL-6, and tumor necrosis factor, resulting in metabolic disorders and a decrease in skeletal muscle mass, making these patients more prone to frailty [29,30]. Frailty and comorbidities have a bidirectional relationship. Chronic inflammation, immune impairment, and abnormal neuroendocrine regulation can increase frailty in patients with comorbidities, whereas frail patients with persistent dysfunction are more susceptible to other diseases. Wu et al. [31] investigated whether there is a correlation between frailty and the number of comorbidities; however, the results of the study were not conclusive. In addition, although frailty susceptibility can increase with age,

evidence suggests that frailty is the result of the cumulative effects of physiology and the environment, and the value of age as a sensitive indicator of frailty is limited [32].

We found that patients who underwent upper GI surgery/pancreatic surgery fell mostly into the persistent frailty or frailty exacerbation categories. The deep location, large scope, and intraoperative bleeding associated with such surgery may significantly increase the risk of frailty. Preoperative frailty is an independent predictor of postoperative complications [33], but little is known regarding the effect of postoperative complications on frailty. The Clavien–Dindo complication classification system was used to predict the incidence of postoperative new-onset frailty [34]. Patients with Grade II or higher complications had a higher rate of new-onset frailty, and this frailty was longer lasting. Postoperative complications increase the body's vulnerability, and system dysfunction aggravates the original weakness. Abdominal drainage has unique advantages in the assessment and prevention of postoperative complications such as gastrointestinal fistula formation, infection, and hemorrhage. However, Harvey et al. reported that catheter-related complications, unplanned extubation, and a lack of nursing experience related to drainage inevitably harm patients [35]. We also found that the frailty of patients was closely related to the duration of drainage tube placement, which may be associated with the effect of drainage on postoperative rehabilitation exercise and the anxiety level of patients. Anxiety is the main negative emotion observed in hospitalized patients. Patients focus too much on somatic symptoms and have low enthusiasm for and compliance with treatment and rehabilitation exercise, which increases the risk of progressing to a state of aggravated or persistent frailty [36].

**4.3.2. Protective factors for frailty trajectories.** We believe that a high level of education represents a protective factor for frailty, whereas ERAS-based management (early mobilization and early feeding) can help alleviate and even reverse postoperative frailty. Patients with higher educational attainment generally have a higher level of health literacy and are more willing and better able to manage themselves. "Early ambulation and early oral feeding" underlies ERAS protocols. Postoperatively, patients have limited activity, and, consequently, their muscle reserves decline [37]. Early activities can promote the recovery of respiratory, gastrointestinal, and musculoskeletal functions; prevent postoperative complications; and improve frailty. Therefore, patients are recommended to be placed semireclined in bed or perform moderate in-bed activities after awakening; they should also undertake out-of-bed activities one day after surgery and subsequently increase the amount of activity they perform each day. The ERAS guidelines state that early recovery from oral feeding after surgery can promote intestinal motility and intestinal mucosal recovery as well as prevent intestinal flora imbalance and bowel displacement [38]. Clinical recommendations include "the recovery of oral feeding without waiting for intestinal ventilation" and "the recovery of oral feeding according to the patient's wishes after surgery". We found that patients who underwent Grade III surgery had stable preoperative function and strong stress-countering ability; surgery effectively relieved pain, and postoperative frailty improved.

## Conclusion

In this study, we found that postoperative frailty was prevalent in older patients who underwent abdominal surgery involving the digestive system. Four types of trajectories were fitted *via* a longitudinal study, and the effects of advanced age, abnormal BMI, education level, comorbidities, surgery type, abdominal drainage tube indwelling time, postoperative complications, ERAS management, and anxiety on the progression of frailty trajectory categories were discussed. Analyzing frailty trajectories and their related factors is conducive to the early identification of changes in frailty in patients undergoing major abdominal surgery and provides targeted guidance for clinical nursing decisions.

## Supporting information

**S1 Data. Raw data.**
(DOCX)

## Acknowledgments

The authors would like to thank Xuanwu Hospital, Capital Medical University doctors and nurses for their contributions to this research. We appreciate the patients who participated in the study for their assistance.

## Author contributions

**Data curation:** Jing Guo, Wenshuang Wang, Xiaoxue Zhang.

**Formal analysis:** Jing Guo, Wenshuang Wang, Xiaoxue Zhang.

**Investigation:** Jing Guo, Wenshuang Wang.

**Methodology:** Wenshuang Wang, Xiaoxue Zhang.

**Project administration:** Yulin Zheng, Xinran Wang.

**Supervision:** Xinran Wang.

**Validation:** Yulin Zheng.

**Writing – original draft:** Jing Guo, Wenshuang Wang.

**Writing – review & editing:** Wenshuang Wang, Yulin Zheng, Xinran Wang.

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
