## [Decision Letter · Decision Letter 0]

5 Jun 2025

Dear Dr. Wang,

If applicable, we recommend that you deposit your laboratory protocols in protocols.io to enhance the reproducibility of your results. Protocols.io assigns your protocol its own identifier (DOI) so that it can be cited independently in the future. For instructions see: https://journals.plos.org/plosone/s/submission-guidelines#loc-laboratory-protocols . Additionally, PLOS ONE offers an option for publishing peer-reviewed Lab Protocol articles, which describe protocols hosted on protocols.io. Read more information on sharing protocols at https://plos.org/protocols?utm_medium=editorial-email&utm_source=authorletters&utm_campaign=protocols.

We look forward to receiving your revised manuscript.

Kind regards,

Barry Kweh

Academic Editor

PLOS ONE

**Journal Requirements:**

1. When submitting your revision, we need you to address these additional requirements. Please ensure that your manuscript meets PLOS ONE's style requirements, including those for file naming. The PLOS ONE style templates can be found at https://journals.plos.org/plosone/s/file?id=wjVg/PLOSOne_formatting_sample_main_body.pdf and https://journals.plos.org/plosone/s/file?id=ba62/PLOSOne_formatting_sample_title_authors_affiliations.pdf 2. In the ethics statement in the Methods, you have specified that verbal consent was obtained. Please provide additional details regarding how this consent was documented and witnessed, and state whether this was approved by the IRB. 3. We suggest you thoroughly copyedit your manuscript for language usage, spelling, and grammar. If you do not know anyone who can help you do this, you may wish to consider employing a professional scientific editing service.  The American Journal Experts (AJE) (https://www.aje.com/) is one such service that has extensive experience helping authors meet PLOS guidelines and can provide language editing, translation, manuscript formatting, and figure formatting to ensure your manuscript meets our submission guidelines. Please note that having the manuscript copyedited by AJE or any other editing services does not guarantee selection for peer review or acceptance for publication.  Upon resubmission, please provide the following: The name of the colleague or the details of the professional service that edited your manuscript A copy of your manuscript showing your changes by either highlighting them or using track changes (uploaded as a *supporting information* file) A clean copy of the edited manuscript (uploaded as the new *manuscript* file) 4. Thank you for stating the following in the Acknowledgments Section of your manuscript: The authors would like to thank Xuanwu Hospital, Capital Medical University doctors and nurses for their contributions to this research and the Beijing Science and Technology Commission for providing financial support. We appreciated the patients participated in the study for their assistance. We note that you have provided funding information that is not currently declared in your Funding Statement. However, funding information should not appear in the Acknowledgments section or other areas of your manuscript. We will only publish funding information present in the Funding Statement section of the online submission form. Please remove any funding-related text from the manuscript and let us know how you would like to update your Funding Statement. Currently, your Funding Statement reads as follows: The author(s) received no specific funding for this work.  Please include your amended statements within your cover letter; we will change the online submission form on your behalf. 5. We note that your Data Availability Statement is currently as follows: All relevant data are within the manuscript and its Supporting Information files. Please confirm at this time whether or not your submission contains all raw data required to replicate the results of your study. Authors must share the “minimal data set” for their submission. PLOS defines the minimal data set to consist of the data required to replicate all study findings reported in the article, as well as related metadata and methods (https://journals.plos.org/plosone/s/data-availability#loc-minimal-data-set-definition). For example, authors should submit the following data: - The values behind the means, standard deviations and other measures reported;- The values used to build graphs;- The points extracted from images for analysis. Authors do not need to submit their entire data set if only a portion of the data was used in the reported study. If your submission does not contain these data, please either upload them as Supporting Information files or deposit them to a stable, public repository and provide us with the relevant URLs, DOIs, or accession numbers. For a list of recommended repositories, please see https://journals.plos.org/plosone/s/recommended-repositories. If there are ethical or legal restrictions on sharing a de-identified data set, please explain them in detail (e.g., data contain potentially sensitive information, data are owned by a third-party organization, etc.) and who has imposed them (e.g., an ethics committee). Please also provide contact information for a data access committee, ethics committee, or other institutional body to which data requests may be sent. If data are owned by a third party, please indicate how others may request data access. 6. Please upload a new copy of Figures 3, 4 and 5 as the detail is not clear. Please follow the link for more information: https://blogs.plos.org/plos/2019/06/looking-good-tips-for-creating-your-plos-figures-graphics/ 7. Please remove your figures from within your manuscript file, leaving only the individual TIFF/EPS image files, uploaded separately. These will be automatically included in the reviewers’ PDF.

**Additional Editor Comments:**

The authors need to justify their methodology and presentation of results given table subheadings are often unclear. The decision to define older patients as > 60 years rather than the traditional WHO or UN definition needs to be discussed. Reviewers have also raised concerns regarding internal consistency and the multivariate models which need addressing.

Reviewers' comments:

Reviewer's Responses to Questions

**Comments to the Author**

1. Is the manuscript technically sound, and do the data support the conclusions?

Reviewer #1: Yes

Reviewer #2: Yes

2. Has the statistical analysis been performed appropriately and rigorously?

Reviewer #1: I Don't Know

Reviewer #2: No

3. Have the authors made all data underlying the findings in their manuscript fully available?

Reviewer #1: Yes

Reviewer #2: Yes

4. Is the manuscript presented in an intelligible fashion and written in standard English?

Reviewer #1: Yes

Reviewer #2: Yes

**Reviewer #1: ** Reviewer Comments

General Comments

This study presents a highly original and clinically relevant analysis of dynamic changes in frailty status among older adults undergoing abdominal surgery. The use of latent class growth modeling (LCGM) to classify postoperative frailty trajectories is novel and valuable. Identifying four distinct frailty patterns provides meaningful insight that may support tailored perioperative care strategies. The authors are to be commended for addressing this important and underexplored topic.

However, there are several methodological and presentation-related issues that need to be addressed to improve the clarity, clinical applicability, and transparency of the study.

Major Concerns

1. Study Population Age Definition (Why ≥60 years?)

The rationale for setting the inclusion criterion at age 60 rather than the more commonly used threshold of 65 years is unclear. A justification should be provided, especially given that frailty-related literature often focuses on populations ≥65 years.

2. Albumin and Other Nutritional Markers

The study discusses nutritional status, yet serum albumin, total protein, and prealbumin—commonly used clinical markers—are not included as explanatory variables. Were these data unavailable, or were they excluded for another reason? Please clarify.

3. Surgical Approach Misrepresentation

Table 1 lists “laparoscopy: 100%, laparotomy: 0%,” which is inconsistent with the types of procedures included (e.g., gastrectomy, pancreaticoduodenectomy), many of which are still commonly performed via open surgery. This discrepancy must be clarified. Actual surgical approach data (laparoscopic vs open) should be accurately reported.

4. Poor Table Design and Lack of Definitions

Many of the tables are difficult to interpret. Variable names are inconsistently labeled or undefined (e.g., “income,” “gastric tube”). Some variables appear more than once (e.g., “preoperative gastric tube”), and the definitions of categorical cutoffs (e.g., “bed time,” “drainage”) are not provided. Without clear definitions, clinical interpretation is impaired. Please reformat the tables for clarity and include a variable definitions table or footnotes.

Minor Concerns

1. Placement of Table 3 (Model Fitting Table)

Table 3 presents technical statistics (AIC, BIC, entropy, BLRT) that may not be accessible or meaningful to general surgical readers. Since it does not contribute directly to the clinical narrative, I suggest moving Table 3 to the supplementary material and focusing the main text on clinically interpretable content such as Figure 2–5.

2. “One in Ten” Rule in Multivariable Models

The multivariable analyses shown in Figures 3–5 include many predictors relative to the group sizes (e.g., 9 predictors in the improvement group with only 21 patients). This may violate the "one-in-ten rule" and lead to overfitting. The authors should clarify how variables were selected and consider model simplification or penalized regression techniques if appropriate.

3. Typographical and Formatting Issues

There are minor inconsistencies in table formatting (e.g., alignment, repeated rows) that reduce readability and should be addressed during revision.

**Reviewer #2:**  Dear Editor,

This article explores the trajectory and influencing factors of frailty in patients undergoing gastrointestinal surgery, with a particular focus on the effects of BMI, Charlson comorbidity index score, type of surgery, intraoperative drainage tube retention time, first time to get out of bed after surgery, the time of the first oral feed after surgery, postoperative complications, mobility, nutritional risk, and anxiety on postoperative frailty.

This study provides a contribution to the literature on this topic, offering detailed data and short-term follow-up of a population that has been digestive tract surgery. However, I still have some questions and suggestions:

1. Whether or not chemotherapy is administered after surgery can significantly impact the manifestation of frailty, as certain chemotherapy drugs can adversely affect physical condition and lead to frailty. Why did the author fail to include chemotherapy as an associated factor?

2. The surgical spot will influence the timing of the first postoperative meal and the placement of a peritoneal drainage tube. In other words, there is multicollinearity among these three factors. If the authors include all three factors in the multivariate analysis simultaneously, it will lead to greater errors and increase the instability of parameter estimation. How do the authors explain or avoid this influence?

3. In Figs. 3 and 4, the OR value of ADL is less than 0, while that of NRS2002 and STAI is are greater than 0. In Fig. 5, the above three values are reversed, while other factors such as bed time and complications are all greater than 0. The authors should provide an explanation for these discrepancies in the discussion section.

4. Many of the data presented in the manuscript are inconsistent and confusing:

1) In table 1, the number of smokers, education level, monthly income, and CCI are not equal to the total number (155).

2) In Table 3, the BIC of four-category model is higher than that of three-category model, however the description indicates the opposite.

3) The content described in the results shows “There were 41 patients in class 3, accounting for 26.7% of the total.” However, in Table 2, the frailty was only 33 before surgery (T1).

4) The numbers for the "no frailty" and "frailty improvement" in Table 4 are inconsistent with those presented in the preceding text. Additionally, there is a typographical error in the male column* frailty improvement" in Table 4. Furthermore, there are problems with the data of BMI*frailty exacerbation, Age*persistent frailty, Drinking status*frailty exacerbation, Habitation status*frailty exacerbation, persistent frailty, frailty improvement, Education level*no frailty, Monthly income*persistent frailty, frailty improvement.

5) The data of Surgical spot* frailty exacerbation, frailty improvement Preoperative pain* frailty exacerbation, frailty improvement Preoperative urinary catheter* persistent frailty Postoperative complications* no frailty, persistent frailty in Table 5 are problematic.

6) In the results section, should the second line of the "Frailty trajectories" section have "table2" instead of "table3"?

**Do you want your identity to be public for this peer review?** For information about this choice, including consent withdrawal, please see our Privacy Policy

Reviewer #1: **Yes: ** Shokei Matsumoto

Reviewer #2: No

---

## [Author Response · Author response to Decision Letter 1]

15 Jul 2025

Dear reviewer,

Thank you very much for your comments and professional advice. These opinions help to lmprove academic rigor of our article Based on your suggestion and request,we have made corrected modifications on the revised manuscript. Meanwhile. The manuscript had be reviewed and edited by language services of AJE.We hope that our work can be improved again.Furthermore,we would like to show the details as follows:

Reviewer #1: Reviewer Comments

Major Concerns

1. Study Population Age Definition (Why ≥60 years?)

The rationale for setting the inclusion criterion at age 60 rather than the more commonly used threshold of 65 years is unclear. A justification should be provided, especially given that frailty-related literature often focuses on populations ≥65 years.

Response 1

The-author's answer:The reasons for setting the inclusion criteria for the study population at ≥60 years old are as follows: On the one hand, the WHO defines the age of elderly individuals in developing countries as ≥60 years, and China's definition and regulations for the elderly also specify ≥60 years; additionally, this study considers that frailty is a non-specific state in older adults characterized by increased physical vulnerability and reduced stress resistance due to decreased physiological reserve. Such individuals are more significantly affected by grade III and IV major surgeries and more likely to develop frailty. Therefore, the population was defined as ≥60 years old based on the above reasons.

2. Albumin and Other Nutritional Markers

The study discusses nutritional status, yet serum albumin, total protein, and prealbumin—commonly used clinical markers—are not included as explanatory variables. Were these data unavailable, or were they excluded for another reason? Please clarify.

Response 2

The-author's answer:In the discussion of nutritional status, this study only used the NRS 2002 scale, and commonly used clinical indicators (e.g., serum albumin, total protein, and prealbumin) were not included as explanatory variables. This is primarily due to limitations related to the study location and time: the hospital where the study was conducted did not require all surgical patients to undergo the laboratory tests required by the study between 2023 and 2024, resulting in the study being unable to accurately obtain relevant data for each patient. Therefore, after discussion by the research team, a professional scale for nutritional risk assessment was selected, which also constitutes one of the limitations of the study. Moving forward, we will also carry out multidisciplinary collaboration to further improve the in-depth analysis of nutritional status.

3. Surgical Approach Misrepresentation

Table 1 lists “laparoscopy: 100%, laparotomy: 0%,” which is inconsistent with the types of procedures included (e.g., gastrectomy, pancreaticoduodenectomy), many of which are still commonly performed via open surgery. This discrepancy must be clarified. Actual surgical approach data (laparoscopic vs open) should be accurately reported.

Response 3

The-author's answer:

The We re-investigated the original medical records of the study participants to verify the surgical approach (laparotomy vs. laparoscopy), clarified the inaccurate data in the report, and made corrections (laparoscopy: 66.45% vs. laparotomy: 33.55%).

4. Poor Table Design and Lack of Definitions

Many of the tables are difficult to interpret. Variable names are inconsistently labeled or undefined (e.g., “income,” “gastric tube”). Some variables appear more than once (e.g., “preoperative gastric tube”), and the definitions of categorical cutoffs (e.g., “bed time,” “drainage”) are not provided. Without clear definitions, clinical interpretation is impaired. Please reformat the tables for clarity and include a variable definitions table or footnotes.

Response 4

The-author's answer:Thanks for your suggestion.However, We have rechecked data related to study participants and revised the table content, including standardizing variable names, clarifying their specific definitions, verifying the data, adding footnotes, and other adjustments. Details are presented in the table. And we hope the revised manuscript could be acceptable for you.

Minor Concerns

1. Placement of Table 3 (Model Fitting Table)

Table 3 presents technical statistics (AIC, BIC, entropy, BLRT) that may not be accessible or meaningful to general surgical readers. Since it does not contribute directly to the clinical narrative, I suggest moving Table 3 to the supplementary material and focusing the main text on clinically interpretable content such as Figure 2–5.

The-author's answer:We think this is an excellent suggestion. We have adjusted the position of the table based on your suggestions, so that it is now presented as supplementary material.

2. “One in Ten” Rule in Multivariable Models

The multivariable analyses shown in Figures 3–5 include many predictors relative to the group sizes (e.g., 9 predictors in the improvement group with only 21 patients). This may violate the "one-in-ten rule" and lead to overfitting. The authors should clarify how variables were selected and consider model simplification or penalized regression techniques if appropriate.

The-author's answer:Thank you for your valuable suggestions. In the multivariate analysis, we incorporated the multivariate model analysis based on the results of the univariate analysis. This may violate the "one-in-ten rule" and lead to overfitting. After group discussion, simplifying the model might affect the conclusions of this study. Therefore, we plan to further include more study subjects, expand the sample size, and conduct in-depth analysis of the influencing factors in subsequent research.

3. Typographical and Formatting Issues

There are minor inconsistencies in table formatting (e.g., alignment, repeated rows) that reduce readability and should be addressed during revision.

The-author's answer:Thanks for your careful checks. We are sorry for our carelessness. Based

on your comments, we have revised the typesetting and formatting issues, especially the tables.

Reviewer #2: Dear Editor,

1.Whether or not chemotherapy is administered after surgery can significantly impact the manifestation of frailty, as certain chemotherapy drugs can adversely affect physical condition and lead to frailty. Why did the author fail to include chemotherapy as an associated factor?

Response 1

The-author's answer:This study primarily focused on the impact of abdominal surgery-related factors on frailty in elderly patients, with a follow-up period extending to 1 month postoperatively. Although chemotherapy, as one of the adjuvant therapies for cancer patients, may lead to decline in physical function and frailty, it was not included in the primary analytical dimension after comprehensively considering its limited relevance to the research objective.

2.The surgical spot will influence the timing of the first postoperative meal and the placement of a peritoneal drainage tube. In other words, there is multicollinearity among these three factors. If the authors include all three factors in the multivariate analysis simultaneously, it will lead to greater errors and increase the instability of parameter estimation. How do the authors explain or avoid this influence?

Response 2

The-author's answer:Before conducting multivariate logistic regression analysis, the study first performed univariate analysis on all variables (including surgical site, timing of the first postoperative meal, and placement of peritoneal drainage tube). Independent influencing factors were screened, and only variables with statistical significance (P<0.05) in the univariate analysis were included in the final multivariate model. This screening process effectively eliminates highly correlated variables, avoids simultaneously including factors with multicollinearity, and thereby reduces model errors and the instability of parameter estimation. Additionally, the surgical sites involved in this study included the gastrointestinal tract, hepatobiliary system, and pancreas. In the preliminary literature review, no significant evidence was found to confirm obvious multicollinearity among surgical site, timing of the first postoperative meal, and peritoneal drainage tube placement. Of course, we have deeply reflected on and discussed the issue raised by the expert. This aspect was not considered in the preliminary research, and we will conduct in-depth studies to further explore how to minimize such errors and enhance the stability of parameters.

3.In Figs. 3 and 4, the OR value of ADL is less than 0, while that of NRS2002 and STAI is are greater than 0. In Fig. 5, the above three values are reversed, while other factors such as bed time and complications are all greater than 0. The authors should provide an explanation for these discrepancies in the discussion section.

Response 3

The-author's answer:Thank you for carefully reading the article and putting forward some valuable suggestions. In response to the issues you raised, we conducted a discussion among the research team and revised the errors. The details are as follows:

The We re-reviewed the original data, conducted statistical analysis and statistical testing, and have revised the OR values in Figures 3 and 4: An OR value <1 indicates a protective factor for frailty status (worsening, persistence, improvement); an OR value >1 indicates an exposure factor, and supplementary explanations have been added in the discussion section.

4. Many of the data presented in the manuscript are inconsistent and confusing:

1) In table 1, the number of smokers, education level, monthly income, and CCI are not equal to the total number (155).

2) In Table 3, the BIC of four-category model is higher than that of three-category model, however the description indicates the opposite.

3) The content described in the results shows “There were 41 patients in class 3, accounting for 26.7% of the total.” However, in Table 2, the frailty was only 33 before surgery (T1).

4) The numbers for the "no frailty" and "frailty improvement" in Table 4 are inconsistent with those presented in the preceding text. Additionally, there is a typographical error in the male column* frailty improvement" in Table 4. Furthermore, there are problems with the data of BMI*frailty exacerbation, Age*persistent frailty, Drinking status*frailty exacerbation, Habitation status*frailty exacerbation, persistent frailty, frailty improvement, Education level*no frailty, Monthly income*persistent frailty, frailty improvement.

5) The data of Surgical spot* frailty exacerbation, frailty improvement Preoperative pain* frailty exacerbation, frailty improvement Preoperative urinary catheter* persistent frailty Postoperative complications* no frailty, persistent frailty in Table 5 are problematic.

6) In the results section, should the second line of the "Frailty trajectories" section have "table2" instead of "table3"?

Response 4

The-author's answer:We were really sorry for our careless mistakes.Thank you for yourreminder.

1)We have re-reviewed the original data, revised the values of the number of smokers, education level, monthly income, and Charlson Comorbidity Index (CCI),

2)We have reconducted statistical calculations to adjust the Bayesian Information Criterion (BIC) value of the four-category model.

3)We re-analyzed and recalculated the original data of the study subjects and revised Data in Table 2.

4)Data in Table 4 have been revised.

5)Data in Table 5 have been revised.

6)Table 2 has been revised to Table 3 in "Frailty trajectories" section.

Thank you very much for your attention and time.Look for ward to-hearing from you.

Yours sincerely,

Xinran Wang

2025.7.12

---

## [Editor Report · Decision Letter 1]

27 Jul 2025

Frailty trajectory and its associated factors in older patients undergoing abdominal surgery involving the digestive system:A longitudinal study

PONE-D-25-12651R1

Dear Dr. Wang,

We’re pleased to inform you that your manuscript has been judged scientifically suitable for publication and will be formally accepted for publication once it meets all outstanding technical requirements.

Kind regards,

Barry Kweh

Academic Editor

PLOS ONE

Additional Editor Comments (optional):

The authors have clarified the difference between their univariate and multivariate analysis, updated their tables and provided a more nuanced discussion of the relevant literature.
---

## [Editor Report · Acceptance letter]

PONE-D-25-12651R1

PLOS ONE

Dear Dr. Wang,

I'm pleased to inform you that your manuscript has been deemed suitable for publication in PLOS ONE. Congratulations! Your manuscript is now being handed over to our production team.

Kind regards,

on behalf of

Dr. Barry Kweh

Academic Editor

PLOS ONE